# OpenReview forum: "Do Models Explain Themselves? Counterfactual Simulatability of Natural Language Explanations"
_ICLR.cc/2024/Conference — Submitted to ICLR 2024_

### Official Review · Reviewer_Sh5n · 2023-10-21

**Soundness:** 2 fair
**Presentation:** 2 fair
**Contribution:** 2 fair
**Rating:** 6
**Confidence:** 4

**Summary:**

The authors investigate the properties of explanations provided by LLMs. Specifically, they investigate a) Whether explanations are `precise`, i.e. can simulate how the model would behave under counterfactual questions b) are `general`, whether they help with the simulatability of queries that are different than the original query. The authors provide a methodology to evaluate explanations for these two criteria and evaluate two LLMs (GPT-3, 4) in two tasks (StrategyQA, SHP).

**Strengths:**

I find that the paper has an interesting object to evaluate (explanations provided by LLMs) and a well-constructed methodology to do so. It appears to be a good addition to the methodologies being developed to evaluate LLM explanations, such as ones provided in the paper (Turpin et al., Creswell et al.).

More specifically,

1. The idea that the quality of an expectation could be measured by the simulatability of counterfactuals is a useful one. The ability of explanations to help humans construct mental models is one that is discussed in the existing literature – however, instantiation in the context of language models appears to be an interesting idea.

2. Formalizing simulatability using logical entailment is again a useful idea. I’m not sure if this exists in the literature – but could be a good way to automate explanation evaluations in general.

3.  I appreciate the sanity checks in Section 5.1 to justify several design choices made by the authors, before moving on to more complex experiments. This clarifies several questions in the mind of a potential reader.

**Weaknesses:**

1. There is a human study in the paper that does not mention whether the study has an IRB approval. Citing ICLR Code of Ethics `Where human subjects are involved in the research process (e.g., in direct experiments, or as annotators), the need for ethical approvals from an appropriate ethical review board should be assessed and reported. ` **How to address: I recommend the authors to apply for an IRB approval from their respective institution to resolve this concern. Ideally this should have done prior to data collection, but I am leaving this to the judgment of Ethics Reviewers / ACs.**

2. One of the conceptual contributions authors make is the proposed criteria to evaluate explanations. However, I’m a bit concerned with both desiderata (generality and precision) provided by the authors:
- 2.1 - The terminological choice of precision looks very much like faithfulness (See Q1), which created significant confusion for me. The definition of precision in Section 3.2 is referred to as faithfulness in many other contexts (as in the cited related works). Unless I’m missing something here, I’m not sure why one needs to invent a new name for it.
- 2.2 - I find the claim `an ideal explanation .. should be generalizable, … it should also reveal how well the model reasons on unseen outputs` unjustified. Why do we really want generalizable claims? Why is it a non-ideal thing to only explain the answer to the question one is responding to? As long as the explanation is faithful, I cannot say being more general or more specific is preferable in this context. **How to address: I’d be curious to hear authors’ justifications for a) Why Precision is not simply Faithfulness? b) How do we claim that we want generality in these explanations?**

3. I am also unclear about what conclusions we can reliably draw from the experiments, or how these may benefit the users/consumers of the explanations.
- One general conclusion appears to be around a more capable model (GPT-4) providing `better`(under the criteria in the paper) explanations than a less capable model (GPT-3).
- However, beyond the relative inferences, I cannot understand whether these numbers are good or not in an absolute sense, thus whether explanations are good or not (Q3). In general, a lack of comparison to interpretable numbers makes the results hard to interpret.  **How to address: One way would be to provide baseline explanations. For instance, one can ask the same set of questions to a set of human users, let the human users provide explanations, and compare those explanations with the ones provided by the models.**

Minor:

1. Imho, the title is not precise. “Do models explain themselves” is pretty generic with the use of the word “model”, but really the authors are focusing on a specific model class, which is autoregressive language models. I’d personally err on the side of precision.

**Questions:**

1. I’m slightly confused about the proposed terminology here. The authors argue that precision is that `they should lead to mental models that are consistent with the model’s behavior.`; this definition sounds quite a bit like faithfulness, in my opinion (also in the related works authors refer to faithfulness as `It is different from faithfulness, which measures whether an explanation is consistent with the model’s own decision process`).

In that, this argues explanation should be faithful to the model’s behavior. Personally, I find precision to have a stronger connotation around explaining what could be explained, and nothing beyond it. I’d love to hear the authors’ opinions about why this is not faithfulness and is precision.

2. The authors empirically argue that generality and precision do not seem correlated -- however I do not necessarily agree with this, and I’m unclear about whether an explanation has to be general as long as it is precise. To be more concrete, one could make an extremely local explanation (i.e. hard to find BLT because markets do not sell BLT) that is extremely precise. For instance, using the formalization in the paper, we could have $|C|$ large but $|C^*|$ small, potentially $0$ or $1$, e.g. if the model uses the answer to explain the answer. Are these two desiderata independent? If they are not, I believe the paper would also benefit from a discussion around this, and also further discussion in Section 5.2.2 where the authors suggest independence.

3. I’m not sure how one can claim that 80% is not precise enough (i.e. the claim `explanations have low precision (80% for binary classification)` in the intro); the number seems nontrivial. Why do we expect a larger number? Do we have statistics about how precise human explanations are? Can this number be even larger than for humans’?

4. What is being reported in Table 5? Is it the correlation coefficient? Is it Spearman or Pearson? What are the p-values? Are the numbers statistically significant?

Minor
1. The inline citations are confusing to me, perhaps it would clarify to replace \cite calls with \citep calls to clarify reading.
2. Page 1, “Muslin countries” -> “Muslim countries”
3. Figure 2 says “robot’s answer”; I’m assuming this is the LLMs’ answer? Otherwise, what’s the robot?

**Details Of Ethics Concerns:**

The authors perform human simulations via Mechanical Turk, but I do not see whether the authors have IRB approvals or exemptions for the human subject experiment.

**Rebuttal:** Answered by the authors, they suggest they do have an IRB.

---

> ### Author Response · Authors · 2023-11-16
> **Response to your review (part 1/2)**
>
> Thanks so much for your questions!
>
> &nbsp;
>
> Q: Did you apply for IRB?
>
> A: Yes we did! Our IRB was approved on 05/09/2023 prior to data collection. We did not submit the certificate due to anonymity constraints, but will include it if the paper is accepted.
>
> &nbsp;
>
> Q: How to understand the precision numbers “in an absolute sense”? Are human-written explanations more precise?
>
> A: Following your advice, we __ran a new evaluation__ to measure the precision of human-written explanations. Just like how we scored LLM explanations, we asked a human annotator to write explanations for the same set of questions, used GPT-4 to generate counterfactual questions, asked the human annotator to answer the counterfactual questions, and scored how often the human annotator’s answer to the counterfactuals is consistent with the simulator’s answer. __Human-written explanations achieved a simulation precision of 95.0, much higher than the precision of LLM-generated explanations (80.6) with p-value < 0.001__. We’ve added this result to the pdf (Section 5.2.1 last paragraph).
>
> &nbsp;
>
> Q: Why is 80% precision “not precise enough”?
>
> A: Explanations are especially useful in high-stakes domains, where 80% precision is insufficient. Plus, human-written explanations have a precision of 95.0, indicating there could be a large room for improvement for LLM generations.
>
> &nbsp;
>
> Q: Why do we want generalizable explanations? Why is an explanation “non-ideal” if it “only explains the answer to the question one is responding to”?
>
> A: When humans form mental models of how a model behaves, generalizability is important so that they can correctly infer the model’s output on examples where they do not see the model’s explanation. For example, a general explanation “birds can fly” enables humans to infer the model’s output on all questions asking whether each kind of bird can fly (e.g., eagles, sparrows, etc.). That way humans do not have to read the model’s explanation for each individual bird one by one, which is tedious and inefficient.
>
> &nbsp;
>
> Q: Intuitively “one could make an extremely local explanation that is extremely precise”. What is reported in Table 5? Is it Spearman or Pearson? Are the numbers statistically significant?
>
> A: We agree with your intuition that an extremely precise explanation can lead to low generality, which motivates our experiments to measure the correlation between generality and precision to see if there is any trade-off (Section 5.2.2 Paragraph 3). Table 5 shows near-zero Spearman correlations between precision and generality, and we observe similar near-zero correlations for Pearson (StrategyQA - BLEU: 0.023, Cosine: 0.003, Jaccard: -0.001; SHP - BLEU: 0.048, Cosine: 0.008, Jaccard: 0.013). We observe statistically significant correlations (p-value < 0.05) only on <3% of the examples where we calculated the correlations. Given these numbers, we conclude that there is no significant trade-off between precision and generality.

---

> > ### Author Response · Authors · 2023-11-16
> > **Response to your review (part 2/2)**
> >
> > Q: How is simulation precision different from “faithfulness”? Why “invent a new name” for simulation precision if it is just faithfulness?
> >
> > A: We did not invent the term “simulation precision”. “Simulation precision” [1,2] and “faithfulness” [3,4,5] are two different terms __defined by prior work__, and our paper __explained their differences in Section 2 Paragraph 2__ and followed the use of their terminology. Both terms measure if an output predictor can guess the model’s output based on its explanation. The difference here is simulation precision requires the output predictor to be humans while faithfulness does not. Because of this difference, raw model weights are faithful because running the model weights as the output predictor yields perfect accuracy, but have zero simulation precision because humans cannot read model weights directly.
> >
> > We focus on simulation precision because explanations need to be human-readable in order for humans to build mental models. Finally, to help clarify on the contribution of our paper, neither counterfactual simulatability or counterfactual faithfulness has been explored before for natural language explanations.
> >
> > &nbsp;
> >
> > Q: There are some issues with the citation format and two typos.
> >
> > A: Thanks for catching these! We corrected the citation format and typos in the pdf.
> >
> > &nbsp;
> >
> > [1] Doshi-Velez, Finale, and Been Kim. "Towards a rigorous science of interpretable machine learning." arXiv preprint arXiv:1702.08608 (2017).
> >
> > [2] Peter Hase and Mohit Bansal. 2020. Evaluating Explainable AI: Which Algorithmic Explanations Help Users Predict Model Behavior?. In Proceedings of the 58th Annual Meeting of the ACL. ACL.
> >
> > [3] Harrington, Leo A., et al., eds. Harvey Friedman's research on the foundations of mathematics. Elsevier, 1985.
> >
> > [4] Marco Tulio Ribeiro, Sameer Singh, and Carlos Guestrin. 2016. "Why Should I Trust You?": Explaining the Predictions of Any Classifier. In Proceedings of the 22nd ACM SIGKDD International Conference on Knowledge Discovery and Data Mining (KDD '16). ACM.
> >
> > [5] Jialin Wu and Raymond Mooney. 2019. Faithful Multimodal Explanation for Visual Question Answering. In Proceedings of the 2019 ACL Workshop BlackboxNLP: Analyzing and Interpreting Neural Networks for NLP. ACL.

---

> > > ### Comment · Reviewer_Sh5n · 2023-11-17
> > > **Response to the rebuttal**
> > >
> > > Thank you for your rebuttal!
> > >
> > > - **IRB**
> > >
> > > Thanks for the clarification! Do you also have the IRB for the latest human evaluation (one where you ask humans to generate explanations to compare precision)? What is the data collection procedure?
> > >
> > > - **Human-written explanations**
> > >
> > > This will be good to have! I have a couple more questions to understand and interpret the new experiment better:
> > >
> > > Could you provide more details about how you recruited the human annotator? Did you recruit through MTurk, if yes why only 1 annotator? Could you also provide a bit more quantitative details about the experiment? Specifically: What is the number of explanations/questions you asked from the human annotator to write?
> > >
> > > Personally, I do not agree with the statement *Explanations are especially useful in high-stakes domains, where 80% precision is insufficient*. In particular, i) yes it seems to have a face value, but ii) in this case neither the authors are evaluating high-stakes settings, nor 80% is clearly good enough or not for such settings. However, I do agree with a statement that would compare against the human baseline and claim there is a gap to close, given that we agree the number 95% is computed reasonably. I will revise my thoughts around this once I have more details about the experiment.
> > >
> > > - **Table 5, Precision and Generality**
> > >
> > > I appreciate the clarification. It would be good to indicate in the caption of the table what the numbers are. If I'm not missing it, in the paper you do not mention what kind of correlation coefficients you are computing (although you did explain it to me in your rebuttal text).
> > >
> > > One point I'm still not convinced about is the authors' argument that "there is no significant tradeoff". e.g. in the specific example regime I gave where we have $|C|$ large but $|C^*|$ small, there is clearly a tradeoff (or isn't there?). While a marginal analysis between the two properties may not reveal a relation, there may be a correlation in certain regimes if we look at conditional distributions, which disappear when averaged out.
> > >
> > > - **Faithfulness vs Simulation**
> > >
> > > Thank you for the clarification here, this was my own personal confusion about the existing terminology.
> > >
> > > **Overall**, thank you again for your clarifications. I will revisit my score once the remaining points are clarified.

---

> > > > ### Author Response · Authors · 2023-11-19
> > > > **Response to the review**
> > > >
> > > > - IRB & Human-written Explanations
> > > >
> > > > Q: Do you also have the IRB for the latest human evaluation where you ask humans to generate explanations to compare precision?
> > > >
> > > > A: Our earlier IRB also covers experiments that compare LLM explanations to human explanations.
> > > >
> > > > &nbsp;
> > > >
> > > > Q: How many annotators did you recruit? How did you recruit them? MTurk? What is the number of explanations/questions you asked from the human annotator to write?
> > > >
> > > > A: Over the past few days, we recruited 8 annotators and annotated a total of 104 explanations. To gather reliable annotations in the limited time window of the author response period, we asked students from our institute rather than MTurks to annotate.
> > > >
> > > > Here is the updated result: human-written explanations have precision 91.5, and GPT-4 explanations have precision 82.8 (p-value < 0.001). We did not evaluate more human-written explanations because of 1) the time-constraint: our evaluation requires the human annotators to also answer multiple counterfactuals for each explanation, 2) statistical significance: we already see a significant difference between human-written explanations and GPT-4 explanations.
> > > >
> > > > &nbsp;
> > > >
> > > > - Precision and Generality
> > > >
> > > > Q: It would be good to indicate in the caption of Table 5 what the numbers are.
> > > >
> > > > A: We have updated the caption of the table in the pdf.
> > > >
> > > > &nbsp;
> > > >
> > > > Q: One point I'm still not convinced about is the authors' argument that "there is no significant tradeoff". e.g. in the specific example regime I gave where we have $|C|$ large but $|C^*|$ small, there is clearly a tradeoff (or isn't there?).
> > > >
> > > > A: Thanks for your question! We will make our claim clearer. We did not intend to claim a general property of the metrics -- we only present an empirical observation on a specific set of explanations (generated by LLMs in our case). The correlation will be different on different explanation sets, so our "no correlation" conclusion is restricted to the explanations generated by current LLMs.

---

> > > > > ### Comment · Reviewer_Sh5n · 2023-11-19
> > > > > **Thank you**
> > > > >
> > > > > Thank you for your clarifications, I appreciate the effort you put into your rebuttal, the new experiments, and presentation improvements. I will be adjusting my score accordingly.

---

### Official Review · Reviewer_wXPv · 2023-10-30

**Soundness:** 3 good
**Presentation:** 3 good
**Contribution:** 2 fair
**Rating:** 6
**Confidence:** 4

**Summary:**

This paper introduces the concept of counterfactual simulatability to evaluate natural language explanations generated by large language models (LLMs). The authors propose two metrics, simulation generality and simulation precision, to measure the ability of an explanation to enable humans to infer the model's outputs on diverse counterfactuals. They evaluate state-of-the-art LLMs on multi-hop factual reasoning and reward modeling tasks and find that the explanations have low precision and do not correlate with plausibility. The authors suggest that naively optimizing for human approval may not be sufficient to improve the quality of explanations. The paper emphasizes the importance of building explanations that help humans build accurate mental models of model behavior.

**Strengths:**

1) This work introduces novel metrics to evaluate the quality of generated explanations by LLMs, with a focus on whether those explanations help humans build mental models of those LLMs.

2) The paper is well written, with clear reasoning and explanations. The methodology and results are clear and easy to follow.

3) The paper addresses an important issue on explainable generations. The proposed evaluation framework provides a valuable tool for assessing the quality of explanations.

4) The experiments are conducted on two different tasks, multi-hop factual reasoning and reward modeling, and the results demonstrate the limitations of existing explanation methods and the need for improvement.

5) The paper's findings and insights have the potential to shape future research and development in the field.

**Weaknesses:**

1) The paper does not compare the proposed metrics with other existing evaluation metrics for explanations. Without such comparisons, it is challenging to determine how the proposed metrics perform in relation to other approaches.

2) The evaluation of counterfactual simulatability is limited to classification tasks, and there is a need to extend it to more complex generation tasks.

3) The human simulation task is complex and subjective, leading to only moderate agreement among human annotators, raising concerns about the reliability of human evaluation. That's extended to LLMs as well.

4) The study assumes that language models have some form of "knowledge", which is not actually the case. Language models don't "know" or "understand" information in the way humans do - they generate responses based on patterns they've learned from training data.

5) Some citations do not follow the formatting instructions (When the authors or the publication are included in the sentence, the citation should not be in parenthesis using \citet{}... Otherwise, the citation should be in parenthesis using \citep{})

**Questions:**

1) The discussion of LLM simulators as proxies for human simulators is interesting, but it would be beneficial to provide more insights into the limitations of using LLMs in this role. What are the potential shortcomings or biases that LLM simulators may introduce? How well do they capture the full range of human reasoning and decision-making?

2) Could you elaborate more on the "Forced" strategy you used for the sanity checks?

3) Could the authors elaborate on how the counterfactual questions were generated? Did they follow specific patterns or were they entirely randomly created?

4) Your evaluation method assumes that the knowledge contained in the explanation is the only information used by the model to make its decisions. How would your method account for scenarios where the model uses additional information not contained in the explanation to make its decision?

---

> ### Author Response · Authors · 2023-11-16
> **Response to your review (part 1/2)**
>
> Thanks so much for your questions!
>
> &nbsp;
>
> Q: Did you “compare the proposed metrics with existing evaluation metrics for explanations”?
>
> A: Yes, we did! We compared our metrics with plausibility, which is the most widely used metric for explanations [1,2,3] (Section 5.2.2 Paragraph 2). We tested the correlation between our simulation precision metric and the plausibility metric to study if our simulation precision metric is already covered by plausibility. We only observed a very weak correlation of +0.012 (Pearson) and +0.021 (Spearman) between simulation precision and plausibility, and thus concluded that the new counterfactual precision metric we proposed is orthogonal to plausibility.
>
> &nbsp;
>
> Q: The human simulation task is complex and subjective, “leading to only moderate agreement among human annotators”. Is the human evaluation still “reliable”? Could you elaborate more on the "Forced" strategy you used for the sanity checks?
>
> A: To address the moderate agreement rate, we ran a sanity check experiment in Section 5.1 to check whether our evaluation procedure can distinguish between explanation systems of different quality under the simulation noise. We constructed a baseline system “Forced” where we forced the model to generate a Post-Hoc explanation conditioned on the answer it assigns a lower score to, and compared its precision score to the “Normal” system where the model generates a Post-Hoc explanation conditioned on the answer it assigns a higher score to. “Normal” outperforms “Forced” significantly by +45.2 precision points (38.2 vs. 83.4; p-value < 1e-16), which verifies that our evaluation procedure can discriminate between explanation systems of different quality despite the simulation noise.
>
> Regarding the fact that the IAA is only moderate: prior work also observed that __simulation is known to be inherently subjective__, and claimed that “variance in explanation ratings is quite high, relative to their scale” when they evaluated simulatability with humans [4]. However, despite the noisy simulation, our work and [4] are still able to __draw reliable conclusions that are statistically significant__ even from noisy human evaluation.
>
> &nbsp;
>
> Q: How would your metric account for scenarios where the model uses additional information not contained in the explanation to make its decision?
>
> A: It depends on what information of the decision process is missing in the explanation. Explanations more specific than the model’s decision process (e.g., the model explains “animals with wings can fly” but uses the decision process “all animals can fly”) are penalized with a lower generality score. Conversely, explanations more generic than the decision process (e.g., the model uses the decision process “animals with wings can fly” but the explanation misses the modifier “with wings”) are penalized with a lower precision score.
>
> &nbsp;
>
> Q: Could the authors elaborate on how the counterfactual questions were generated?
>
> A: We use LLMs to generate counterfactuals (Figure 2). To generate counterfactuals for each explanation from the QA model, we show the generator LLM the explanation and prompt it to generate ten follow-up questions for which it can confidently guess the QA model’s answer based on the shown explanation. We show the prompts we use in Appendix B. We show that our LLM-based generator outperforms the PolyJuice baseline on diversity (Section 5.1 Paragraph 4).
>
> &nbsp;
>
> [1] Herman, Bernease. "The promise and peril of human evaluation for model interpretability." arXiv preprint arXiv:1711.07414 (2017).
>
> [2] Lage, Isaac, et al. "An evaluation of the human-interpretability of explanation." arXiv preprint arXiv:1902.00006 (2019).
>
> [3] Jacovi, Alon and Yoav Goldberg. “Towards Faithfully Interpretable NLP Systems: How Should We Define and Evaluate Faithfulness?” Annual Meeting of the Association for Computational Linguistics (2020).
>
> [4] Hase, Peter, and Mohit Bansal. "Evaluating explainable AI: Which algorithmic explanations help users predict model behavior?." arXiv preprint arXiv:2005.01831 (2020).

---

> ### Author Response · Authors · 2023-11-16
> **Response to your review (part 2/2)**
>
> Q: How can we evaluate counterfactual simulatability on “more complex generation tasks”?
>
> A: That’s an important but also challenging research question,  our paper discussed possible solutions in Section 6 Paragraph 1 as future work. The key difference between classification tasks and generation tasks is that multiple answers can be correct in generation tasks, so it is harder to define what it means for a human to “correctly” simulate the model’s output. One possible solution we proposed in the paper is contrastive simulation [5,6,7], where a human simulator is shown the model’s output mixed with fake outputs (distractors) and selects which output is from the model based on the model’s explanation.
>
> &nbsp;
>
> Q: The discussion of LLM simulators as proxies for human simulators is interesting, but it would be beneficial to provide more insights into the limitations of using LLMs in this role.
>
> A: Inspired by [8], one possible future direction is to study if GPT-4 can simulate each annotator even better when given a few annotations from that annotator to learn annotator-specific pattern/bias. Using one LLM simulator for each individual and aggregating the outputs from different simulators may further improve IAA.
>
> &nbsp;
>
> Q: The study assumes that language models have some form of "knowledge", which is not actually the case.
>
> A: We do not assume that LLMs have any form of “knowledge”. Our evaluation procedure is behavioral, which only relies on examining the input-output behavior of the model.
>
> &nbsp;
>
> Q: Some citations do not follow the formatting instructions.
>
> A: Thanks for pointing this out! We have corrected the citation format in the pdf.
>
> &nbsp;
>
> [5] Jacovi, Alon et al. “Contrastive Explanations for Model Interpretability.” EMNLP 2021.
>
> [6] Miller, Tim. “Contrastive Explanation: a Structural-Model Approach.” The Knowledge Engineering Review, vol. 36, 2021, p. e14., doi:10.1017/S0269888921000102.
>
> [7] Kayo Yin and Graham Neubig. 2022. Interpreting Language Models with Contrastive Explanations. EMNLP 2022.
>
> [8] Aher, Gati, Rosa I. Arriaga, and Adam Tauman Kalai. “Using large language models to simulate multiple humans.” arXiv preprint arXiv:2208.10264 (2022).

---

> ### Author Response · Authors · 2023-11-22
> **Questions or comments for our rebuttal/paper?**
>
> As we are reaching the end of the rebuttal period, please let us know if you have any further questions or comments about our paper/rebuttal. Thanks again for reviewing our paper!

---

> > ### Comment · Reviewer_wXPv · 2023-11-22
> > **Thanks.**
> >
> > Thank you for your thorough and insightful clarifications in response to the review. I appreciate the significant effort you've invested in addressing the concerns raised and enhancing the presentation of your findings.

---

### Official Review · Reviewer_yudN · 2023-11-01

**Soundness:** 2 fair
**Presentation:** 3 good
**Contribution:** 2 fair
**Rating:** 5
**Confidence:** 4

**Summary:**

The paper proposes counterfactual simulatability, which pertains to whether a human can predict a model's response to several questions when given its previous answer and explanation of a similar question. Due to the disadvantages of human evaluation, the authors leverage GPT-4 and GPT-3 for counterfactual generation and GPT-4 for imitating a human simulator. By leveraging this pipeline, they claim that they discover the defects of LLMs on simulatiprecision and the weak correlation of simulation precision and plausibility, which drive them to draw the conclusion that RLHF may not improve counterfactual simulatability.

**Strengths:**

1. The topic of the paper, which can be potentially applied to solving the hallucination of LLM, is interesting and important to the current NLP field.
2. The analysis of the experiments is abundant.
3. The paper is clearly written and easy to follow.

**Weaknesses:**

1. Whether GPT-4 can approximate human simulators remains a doubt. As demonstrated in Table 2, the authors state that the numbers reported under the columns of  H-GPT3 and H-GPT4 are calculated by the exact IAA with humans devided by the average IAA between humans. Therefore, the number is a percentage and the true number is much lower, which means that GPT-4 has similar agreement with humans as humans do with each other is not convincing.
2. Within the same context, if GPT-4 exhibits a level of agreement with humans akin to that among humans themselves, it does not inherently establish its competence as an accurate representation of human cognition. An effective approximation should inherently exhibit a strong alignment with human behavioral patterns. In simpler terms, the findings merely attest to GPT-4's ability to replicate the same range of diversity observed among humans in their interactions with one another.
3. The paper's underlying motivation is not explicitly articulated. The rationale for assessing such a metric remains obscure, as it exhibits characteristics reminiscent of the phenomenon of "hallucination." Consequently, the apparent issue appears to revolve around the occurrence of hallucinatory responses to similar questions, which fails to underscore its novelty or distinctive contribution to the field.
4. The conclusion of Table 3 is not correct. As shown in the table, GPT-mix demonstrates a higher similarity score compared with other models, therefore the conclusion should be GPT-mix generates more relevant counterfactuals but not more diverse ones.
5. Typos: The 2nd paragraph of Section 3.1: "Expectation"  doesn't match with the formula listed below.

**Questions:**

Please refer to weaknesses section for details.

---

> ### Author Response · Authors · 2023-11-16
> **Response to your review**
>
> Thanks so much for your questions!
>
> &nbsp;
>
> Q: How is your simulation precision metric related to the phenomenon of “hallucination”?
>
> A: In the literature, “hallucination” is commonly defined as “generation of wrong facts” [1,2,3,4]. Our simulation precision metric does not measure hallucination -- it measures if the explanation __reflects the true decision process of the model__. Below is an example where the explanation is not a hallucination but not precise either. If a model answers “yes” to the question “are chickens warm-blooded?” and explains that “all birds are warm-blooded”, but answers “no” to the question “are sparrows warm-blooded?”, then 1) this explanation is not a hallucination, since it is factually correct that “all birds are warm-blooded, but 2) this explanation is not precise, in that it does not reflect the true decision process of the model on counterfactuals.
>
> &nbsp;
>
> Q: Re: GPT-4 as a proxy of human simulators, even though “GPT-4 exhibits a level of agreement with humans akin to that among humans themselves”, did you check if GPT-4 “exhibit[s] a strong alignment with human behavioral patterns”?
>
> A: Following your advice, we conducted __an additional evaluation__ to test if GPT-4 has similar behavioral patterns as human simulators. Specifically, we studied if GPT-4 has higher agreement with humans on counterfactuals where human-human agreement is high. We measured the correlation between human-GPT-4 agreement and human-human agreement across 1532 counterfactual questions, and observed a strong correlation of Pearson coefficient +0.398 (p-value = 1e-58). This result shows that the GPT-4 simulator has some similar behavioral patterns as human simulators. We’ve added this result to the pdf (Section 5.1 Paragraph 3).
>
> To strike a balance between scientific rigor and economic feasibility, our work __followed the practice from the prior literature__ to use LLMs for automatic evaluation [5,6,7], supported by the fact that LLMs do exhibit similar behavioral patterns as humans in many situations [8,9,10]; in fact, __we consider it necessary to use automatic evaluation to maintain rigor as well__, since using GPT-4 as the simulator for part of the experiments is necessary in order to be able to do all the robustness checks (e.g., making sure that our metric is sensible and can distinguish between good and poor explanation systems in Section 5.1 Paragraph 2) given a fixed budget.
>
> Finally, our paper also __used real actual human subjects__ for all experiments on StrategyQA (Section 5.2) and showed that all conclusions are equally valid.
>
> &nbsp;
>
> Q: Table 3 shows that GPT-mix “demonstrates a higher similarity score compared with other models”, so the conclusion should be that GPT-mix does not generate “more diverse ones”.
>
> A: We’d like to clarify some potential misunderstandings. The numbers in Table 3 are diversity scores, not similarity scores. We define diversity in Section 3.1 as one minus the expected similarity between any two simulatable counterfactuals, so Table 3 shows that GPT-mix generates more diverse ones.
>
> &nbsp;
>
> Q: Typo: The “expectation” in 2nd paragraph of Section 3.1 doesn’t match with the formula listed below.
>
> A: Good catch! The expectation formula should state that x’ \neq x’’. We have updated this in the pdf.
>
> &nbsp;
>
> [1] Dhuliawala, Shehzaad, et al. “Chain-of-verification reduces hallucination in large language models.” arXiv preprint arXiv:2309.11495 (2023).
>
> [2] Kurt Shuster, Spencer Poff, Moya Chen, Douwe Kiela, and Jason Weston. 2021. Retrieval Augmentation Reduces Hallucination in Conversation. In Findings of the Association for Computational Linguistics: EMNLP 2021. ACL.
>
> [3] Zhang, Muru, et al. “How language model hallucinations can snowball.” arXiv preprint arXiv:2305.13534 (2023).
>
> [4] Manakul, Potsawee, Adian Liusie, and Mark JF Gales. “Selfcheckgpt: Zero-resource black-box hallucination detection for generative large language models.” arXiv preprint arXiv:2303.08896 (2023).
>
> [5] Liu, Yixin, et al. “On Learning to Summarize with Large Language Models as References.” arXiv preprint arXiv:2305.14239 (2023).
>
> [6] Fu, Jinlan, et al. “Gptscore: Evaluate as you desire.” arXiv preprint arXiv:2302.04166 (2023).
>
> [7] Liu, Yang, et al. “Gpteval: Nlg evaluation using gpt-4 with better human alignment.” arXiv preprint arXiv:2303.16634 (2023).
>
> [8] Huijzer, Rik, and Yannick Hill. “Large Language Models Show Human Behavior.” PsyArXiv, 31 Jan. 2023. Web.
>
> [9] Binz, Marcel, and Eric Schulz. “Turning large language models into cognitive models.” arXiv preprint arXiv:2306.03917 (2023).
>
> [10] Aher, Gati, Rosa I. Arriaga, and Adam Tauman Kalai. “Using large language models to simulate multiple humans.” arXiv preprint arXiv:2208.10264 (2022).

---

> ### Author Response · Authors · 2023-11-22
> **Questions or comments for our rebuttal/paper?**
>
> As we are reaching the end of the rebuttal period, please let us know if you have any further questions or comments about our paper/rebuttal. Thanks again for reviewing our paper!

---

> > ### Comment · Reviewer_yudN · 2023-11-23
> > **Not all concerns are well addressed**
> >
> > Dear Authors,
> >
> > Thank you for the effort in providing a rebuttal. However, some of my proposed concerns are not even touched, e.g., **Weakness 3**. Moreover, the Q1 may not be interpreted correctly. The concern lies in the appropriateness of computing the simulation appropriateness metric. There is also no indication in the revision that the detailed answer to Q4 will be well-reported in the final version. Overall, I will keep my original score and tend to reject this work.

---

### Meta-Review · Area_Chair_1uhF · 2023-12-09

**Metareview:**

This paper evaluates counterfactual simulatability of natural language explanations generated by LLMs, i.e. the extent to which a human can predict the model’s response to novel inputs. The paper proposes measures for “simulation precision” and “simulation generality”, and presents experiments with GPT-3 and 4, on multihop factual reasoning (StrategyQA), and reward modeling (Stanford Human Preference). The results show that LLM generated explanations do not perform well with respect to the above metrics. In most analyses, GPT models are used as a replacement for humans in counterfactual generation and measuring simulatability.

The topic of this paper is timely and important. In addition, the reviewers found the experiments thorough and the draft well-written. Casting simulatability as logical entailment is an interesting and useful framing that this paper proposes.

Over the discussion period, authors have made several edits that have improved the clarity of the presentation, added missing details, and started to address bigger concerns. For example, whether using GPT4 as a replacement for humans is reasonable, in particular, if “explanations” are ultimately to aid humans in understanding the model’s behavior. Authors have referred to prior work to justify this choice. There have been concerns about lack of proper baselines to be able to interpret the metrics. Authors have added a new pilot user study with human raters that better contextualize simulatability baselines for binary classification tasks. These are great steps to address these concerns, but not enough evidence to justify use of GPT-4 as a comparable choice for both counterfactual generation, and for simulatability measurement. In addition, this work can benefit from better positioning with respect to prior work. The need for introducing yet another metric for evaluating explanations is not well-justified. Some of the confusions also arise from inconsistencies, or defining metrics that actually measure something else (e.g. interchangeable use of diversity and generality). Precision and generality are conceptually not expected to be positively correlated, but rather complementary, as highlighted by one of the reviews. Measuring the correlation is not the right analysis to capture the trade-off between the two. It is unclear what the hypothesis being tested in 5.2.2 is, and whether it is reasonable.

Overall, the paper has lots of potential. But it can significantly benefit from another round of revision to improve positioning of the work and measurement metrics, and better justifying full automation, especially in the context of explanations.

**Justification For Why Not Higher Score:**

- Needs stronger justification for full automation to measure the quality of explanations. After all, explanations are supposed to help humans understand the model. There is some discussion about the connection to hallucination over the rebuttal period, perhaps as an attempt to motivate the fully automated approach taken in the paper. However, there is no mention of that in the manuscript.
- Positioning with respect to prior work needs improvement. Introducing new metrics is not well-justified, and can lead to confusion, and perhaps not the right metric for what they claim to measure. There is inconsistent use of terminology that exacerbates the confusion.

**Justification For Why Not Lower Score:**

N/A

---

### Decision · Program_Chairs · 2024-01-16

Reject